# Invasive Candidiasis in the Intensive Care Unit: Where Are We Now?

**DOI:** 10.3390/jof11040258

**Published:** 2025-03-27

**Authors:** Jose A. Vazquez, Lissette Whitaker, Ana Zubovskaia

**Affiliations:** Division of Infectious Disease, Medical College of Georgia, Augusta University, 1120 15th Street, Augusta, GA 30912, USA; lwhitaker@augusta.edu (L.W.); azubovskaia@augusta.edu (A.Z.)

**Keywords:** *Candida*, candidemia, invasive candidiasis, antifungals

## Abstract

Invasive fungal infections in the intensive care unit (ICU) are not uncommon and most cases are caused by *Candida* species, specifically *Candida albicans*. However, recently, there has been an increase in non-*albicans Candida* spp. (*C. glabrata*; *C. parapsilosis*) causing invasive fungal infections. This has led to an increasing awareness of this infection due to the increase in documented antifungal resistance in many of these *Candida* species. In addition, manifestations of invasive candidiasis are often non-specific, and the diagnosis remains extremely challenging. Unfortunately, delays in antifungal therapy continue to hamper the morbidity; length of stay; and the mortality of these infections. Although the echinocandins are the drugs of choice in these infections, antifungal resistance among the non-*albicans* species (*C. glabrata*; *C. krusei*; *C. auris*; *C. parapsilosis*) is being observed more frequently. This has led to an increase in morbidity and mortality, specifically in critically ill patients. Overall, the diagnosis and management of invasive candidiasis in the ICU remain challenging. It is imperative that the critical care physician keeps this infection at the forefront of their differential diagnosis in order to decrease the mortality rate of these individuals. In this review, we discuss the current epidemiologic trends, diagnosis, and management of invasive candidiasis in the intensive care unit setting.

## 1. Introduction

Fungal infections continue to be a dominant concern in immunocompromised patients. Recently, the Centers for Disease Control (CDC) and the World Health Organization (WHO) categorized fungal infections into different threat categories [1,2,3]. The published priority pathogens are ranked into three separate categories: critical, high, and medium priority. Of these categories, 6 of the 19 pathogens belong to the genus *Candida* (Table 1).

These encompass a range of infections caused by numerous *Candida* species. The growing problem reflects the enormous increase in the pool of patients at risk and the increased opportunity for *Candida* species to colonize and invade tissues which are normally resistant to invasion. *Candida* species are true opportunistic pathogens that are able to exploit recent advances in medical care and gain access to the circulatory system and invade deeper tissues. As expected, *Candida* species specifically affect high-risk patients who are either critically ill or immunocompromised.

## 2. Epidemiology

Invasive candidiasis (IC) remains the most common cause of invasive fungal infections worldwide. In fact, 6 of the top 19 fungal infections worldwide are caused by *Candida* species. There are over 160 different *Candida* species in nature; however, 10 of these produce over 95% of all candidal infections in humans [4]. These include *C. albicans*, *Nakaseomyces glabrata* (*Candida glabrata*), *C. parapsilosis*, *C. tropicalis*, *C. krusei*, *C. lusitaniae*, *C. guillermondii*, *C*. *dubliniensis,* and *C. auris* (Table 1). However, over the past decade, we have seen a shift from the more susceptible *Candida* species (*C. albicans*) to the less susceptible or resistant non-*albicans Candida* species, especially *C. glabrata* and, more recently, *C. auris* [4,7]. Of these non-*albicans Candida* species, *C. glabrata*, *C. krusei*, *C. lusitaniae*, and *C. auris* are of greatest importance because of their intrinsic resistance or the development of secondary resistance to current antifungals [4,7]. Of these, *C. glabrata* is probably the greatest threat, since it is the second most commonly isolated *Candida* species and has demonstrated increasing resistance to fluconazole (20–40%). Overall, it is intrinsically less susceptible to azoles and polyenes. A second potential threat is *C. auris*, which was first described in 2009 and has been reported in over 25 countries [8,9,10]. It has several unique characteristics including multidrug resistance to all antifungals, including almost 100% resistance to fluconazole, 20–30% resistance to polyenes, and occasional resistance to echinocandins, as well as the ability to colonize patients and persist in the healthcare environment [8,9,10] (Table 2).

Numerous epidemiologic studies have shown that all *Candida* species can be transmitted from person to person and from environment to person [4,8,9,10]. This has been recently described in patients that develop infections due to *C. auris*, whereby environmental colonization by this organism leads to hospital-acquired infections. Prior studies evaluating the molecular epidemiology of candidal infections have also shown environment-to-patient transmission of identical strains of *C. albicans* and *C. glabrata* [11,12].

**Table 2 jof-11-00258-t002:** Relative in vitro susceptibility patterns of the *Candida* species [13,14,15,16,17].

	Fluconazole	Voriconazole	Echinocandins	Polyenes	Ibrexafungerp	Fosmanogepix
*C. albicans*	S	S	S	S	S	S
*C. glabrata*	variable	variable	S	S	S	S
*C. parapsilosis*	S	S	S	S	S	S
*C. tropicalis*	S	S	S	S	S	S
*C. krusei*	R	S	S	S		
*C. lusitaniae*	S	S	S	variable	S	S
*C. auris*	R	R	S	variable	S	S

Clinical and autopsy studies have confirmed the significant increase in the incidence of candidemia and disseminated candidiasis. This increase is multifactorial and reflects both increased recognition as well as a growing population of high-risk patients. Furthermore, the increase in infection also reflects the improved survival of patients with underlying immunosuppression and multiple comorbidities.

Candidemia and disseminated candidiasis mortality rates have not improved much over the past two decades and remain in the 30–40% range. Furthermore, in neutropenic patients, the mortality rate approaches 100% [4]. In addition, the diagnosis of IC continues to be associated with significant increases in length of stay (70 days vs. 40 days) when compared to patients with no evidence of IC.

## 3. Pathogenesis and Host Defenses

In most fungal infections, host defects play a major role in the development of candidiasis. The skin constitutes a highly effective, impermeable barrier to *Candida* spp. Disruption of the skin from wounds, burns, and intravenous catheters permits invasion by opportunistic colonizing organisms. Effective phagocytosis is the critical defense mechanism that prevents candidal deep-tissue invasion, thereby limiting candidemia and preventing disseminated infection [18].

The initial step in the development of candidal infections is colonization of the skin and the gastrointestinal tract. Once the GI tract mucosa is disrupted by either chemotherapy (mucositis), trauma, or sepsis, organisms penetrate the injured tissue and gain access to the lymphatics and the bloodstream [4,19,20].

Numerous studies have demonstrated the most common risk factors contributing to candidemia and disseminated candidiasis. These include central venous catheters, prolonged ICU stays (>3 days), neutropenia, broad-spectrum antimicrobials, total parenteral nutrition, major surgery (GI tract), uncontrolled diabetes mellitus, *Clostridium difficile* infections, solid organ transplants, continuous renal replacement therapy (CRRT), extracorporeal membrane oxygenation (ECMO), the development of acute renal failure with hemodialysis, mechanical ventilation (>3 days), and severe acute pancreatitis [4,5,6,21,22,23] (Table 3). Large multicenter studies examining the incidence of IC in the ICU have been conducted over the past decade. The reported rates of candidemia vary significantly between 3.5 and 16.5 per 1000 admissions. However, due to inter-center variability, the fact that most studies focused on candidemia only, and the fact that some encompassed cases that likely represented colonization rather than IC, evaluation of IC incidence trends in the ICU over time remains a challenge [24,25,26,27].

A subset of postsurgical patients, particularly those with recurrent gastrointestinal perforation, anastomotic leaks, or acute necrotizing pancreatitis, are uniquely at high risk for developing candidemia and IC [28,29]. Several different risk factors have been derived from multiple studies. However, many of these risk factors lack specificity and broad application that would render most ICU patients eligible for empirical antifungal therapy [4,6,22].

A systemic review and meta-analysis by Thomas-Ruddel et al. evaluated 34 different studies evaluating the most important risk factors for candidiasis. The study evaluated 29 possible risk factors for IC in the ICU, including demographic factors, comorbid conditions, and medical interventions [27]. The authors found that the risk factors associated with the highest rate of IC were broad-spectrum antibiotics (OR, 5.6; 95% CI, 3.6–8.8), blood transfusions (OR, 4.9; 95% CI, 1.5–16.3), *Candida* colonization (OR, 4.7; 95% CI, 1.6–14.3), central venous catheters (OR, 4.7; 95% CI, 2.7–8.1), and total parenteral nutrition (OR, 4.6; 95% CI, 3.3–6.3). However, interaction among the various risk factors is probably high (Table 3) [4,6,22].

Within the hospital setting, the departments with the highest rates of candidemia include ICUs, surgical units, trauma units, and neonatal ICUs. Moreover, 25–50% of all nosocomial candidemia occurs in ICUs. More recently, non-*albicans Candida* species have also increased significantly in ICUs due to the increased use of fluconazole.

## 4. Manifestations

Candidal infections can present in a wide spectrum of clinical syndromes depending on the type of infection and the degree of immunosuppression. Unfortunately, there are no characteristic manifestations of IC. It has been suggested by Rex JH et al. and others that candidemia and systemic candidiasis may be divided into four overlapping groups or syndromes: catheter-related candidiasis, acute disseminated candidiasis, chronic disseminated candidiasis, and deep-tissue infection [30,31]. Although hematogenous dissemination occurs at some stage of the evolution of all four syndromes, only the first two are frequently associated with positive blood cultures. Thus, the use of candidemia as a marker of IC results in the underestimation of the true incidence of IC.

The clinical manifestation of candidemia and IC can vary significantly, including fever alone, leukocytosis alone, the absence of any organ-specific manifestations, or a wide spectrum of manifestations, including culture-negative septic shock (Table 4). Frequently, manifestations of candidemia may be superimposed on those of the underlying pathology and are indistinguishable from those associated with bacterial infections [4,6,20,22].

Dissemination to multiple organ systems may occur in association with candidemia, especially to the kidney, liver, spleen, skin (macronodular skin lesions ~10%), eyes (endophthalmitis) (5–10%), brain, CSF, and heart valves [6,22].

## 5. Diagnosis

For the diagnosis of IC, findings from laboratory studies are non-specific and lack sensitivity [32,33]. It is most imperative to realize that the early and rapid detection of IC is important to initiate appropriate antifungal therapy and thus reduce the mortality rate associated with invasive candidiasis [34].

Blood cultures are generally the initial approach to diagnose candidemia and candidiasis. However, even if positive, it can occasionally take several days before *Candida* is identified, species identification is performed, and the in vitro susceptibility data are gathered. Unfortunately, blood cultures are positive only in 40–70% of cases of candidiasis [32,33]. In fact, in a large retrospective autopsy published by Roosen et al. in 2000, the authors evaluated 100 adult patients in the MICU. Of these, 81% of the patients had a diagnosis made antemortem. However, of the diagnoses that were missed antemortem (36%), the only infectious process missed was systemic candidiasis (36%) [35].

In other instances of deep-seated candidiasis, tissue cultures are needed. The incidence of deep-seated candidiasis without concomitant candidemia in the ICU is unclear due to challenges in obtaining sterile specimens for microbiological confirmation. Intra-abdominal candidiasis (IAC) accounts for most deep-seated cases of IC, with ~ 30% occurring in the critical care unit [29,36]. Perforation, anastomotic leaks, repeat laparotomies, necrotizing pancreatitis, and abdominal organ transplants increase risk; therefore, incidence is higher in surgical ICUs [36].

## 6. Blood and Tissue Cultures

The gold standard for diagnosing invasive candidiasis is blood cultures, even though their sensitivity is low and the time required for species identification usually exceeds 48 h. Tissue cultures require a surgical approach and are sometimes hard to perform in an already critically ill patient. However, more recently, interventional radiology with percutaneous abscess drainage has produced dramatic results. To overcome these issues, culture-based tests can be combined with non-culture-based assays [33,37,38] (Table 5). For example, using MALDI-TOF technology, once the organism is isolated, the time to identification can be shortened to less than 2 h.

Approximately 50% of blood cultures remain negative despite a diagnosis of IC and, occasionally, a missed diagnosis of candidemia is also encountered for numerous reasons [32,33]. However, candidemia due to line-related infections or endovascular infections with no deep-seated candidiasis are most likely to result in a positive blood culture. Patients with candidemia plus deep-seated candidiasis may present with occasional positive blood cultures. On the other hand, patients with deep-seated candidiasis, most commonly in the setting of intra-abdominal perforation, rarely have positive blood cultures [32,33].

Recognizing the varied spectrum of *Candida* infections (candidemia vs. candidiasis or both) that clinicians might encounter will help with early treatment and performing an accurate diagnostic approach [4,22].

## 7. Non-Culture-Based Tests (NCBTs)

The potential of NCBTs is increasingly being utilized to assist in clinical decision making in order to guide the following: (1) the *initiation* of pre-emptive antifungal therapy (AFT); (2) the discontinuation or withholding of empirical AFT; (3) the monitoring of clinical improvements in patients with IC. Although they are somewhat sensitive, these tests are frequently non-specific and broad-spectrum. Unfortunately, most institutions do not carry these assays in-house and, thus, it may take several days for the results to be returned. This makes them difficult to use in real time to assess for active infection [42].

1,3-*β*-d-glucan (BDG) test is one of the most widely used NCBTs. The assay has shown promise in the diagnosis and treatment stratification of systemic candidiasis infections in the ICU [22,37,39]. It is important to note that a positive BDG test is suggestive of infection and should be used in combination with clinical manifestations in high-risk groups (Table 5).

In addition to monitoring qualitative BDG test results during interval testing, tracking quantitative values following the initiation of antifungal therapy may be used as a prognostic marker for patient response. However, the BDG test is a pan-fungal assay (it recognizes other fungi besides *Candida* species) with a sensitivity of 85% and a specificity of 40–90% [22,37,39].

A study conducted on 257 patients with proven IC that were treated with anidulafungin concluded that a decrease in BDG levels during therapy was associated with treatment success [40]. Consistently decreasing BDG levels during treatment have been shown by multiple groups to result in a favorable therapeutic response among patients with proven or probable invasive fungal infections.

Clinicians should also be wary of scenarios that may lead to false-positive BDG results. For example, certain medications can falsely elevate beta-D-glucan levels, including intravenous amoxicillin–clavulanate and piperacillin–tazobactam [38]. On occasion, infections by certain bacterial organisms can also lead to false positives including *S. pneumoniae* and *P. aeruginosa*, both of which also produce some degree of BDG. Another clinical scenario to be aware of is in hemodialysis patients since the cellulose filters used in dialysis may release BDG substrates that can lead to false positives [43,44]. Blood levels of BDG can also be elevated in the setting of fungal translocation via the GI tract due to a compromised intestinal epithelial barrier during episodes of mucositis or diarrhea due to immune dysfunction, gut damage, or altered gut microbiota [43,44].

The T2 magnetic resonance (T2MR) *Candida* assay (T2 Biosystems, Lexington, MA 02421, USA). is a rapid diagnostic test that uses magnetic resonance technology to directly detect and identify the presence of the five most common *Candida* species: *C. albicans*, *C. glabrate*, *C. parapsilosis*, *C. tropicalis,* and C. *krusei* [40]. The assay amplifies *Candida* DNA and detects the amplified fragments in whole blood using magnetic resonance properties, allowing for the faster detection of *Candida* in the blood compared to traditional culture methods [41,45,46] [Table 5].

In a study of 1801 subjects, T2MR Candida analyzed whole-blood samples and showed a sensitivity of 91.1% and a specificity of 99.4% with a turnaround time of 3 h [41].

There are several potential benefits when using the T2MR technology including the early diagnosis of candidemia, which can lead to improved therapeutic algorithms in invasive candidiasis. Furthermore, it can lead to increased case detection, species-specific tailored antifungal therapy, and a decrease in empiric antifungal treatment. This assay can play an adjunctive role as a stewardship tool to limit unnecessary antifungal utilization. It has been shown to have a high specificity and NPV when compared to blood cultures.

The BDG assay and T2MR Candida are now commonly used assays in the USA, where they have shown to be very helpful when deciding to discontinue empiric antifungal therapy due to their high NPV [47].

## 8. Multiplex Candida Real-Time PCR

The use of PCR in the diagnosis of fungal infections continues to be challenging. In fact, at the present time, for technical reasons, there are no commercially available tests. In a review of more than 50 standard, nested, or real-time PCRs, the pooled sensitivity and specificity were 95 and 92%, respectively, for invasive candidiasis [48]. Unfortunately, no PCR assays have been approved for commercial use in the US.

This test is expensive to perform, however, and although costs may be balanced by more accurate earlier diagnosis with reduced administration of antifungal agents and reduced mortality rates, further studies are needed to validate its use and assess its cost–benefit ratio [49].

## 9. Microbial Cell-Free DNA (mcfDNA) Sequencing

Microbial cell-free DNA (mcfDNA) represents a new frontier in infectious disease diagnostics that relies on the detection of fragments of pathogen DNA shed into the bloodstream from a distant site of infection. Detection can be achieved noninvasively through a blood draw; thus, the method has been referred to as a “liquid biopsy” [50]. This technique is often used as a final attempt in diagnosing infection, especially in immunocompromised and pediatric populations. Unlike traditional microbiological diagnostics, mcfDNA sequencing does not rely on organism growth and to some extent can detect a signal despite the presence of anti-infective treatments. In addition, it also allows for species identification across multiple kingdoms, including bacterial, viral (DNA viruses only), fungal, or parasitic organisms, without requiring prior knowledge of the specific organism [50]. This cell-free DNA (cfDNA) detection in plasma is a novel testing modality for the noninvasive diagnosis of invasive fungal infections (IFIs).

There are several limitations to these sensitive assays, including the leakage of molecular material from the microbiome that can be absorbed into the bloodstream and lead to a false positive. This is especially true in patients with acute inflammation, such as pneumonia. The detection of bacterial DNA in plasma might be particularly problematic in patients who are immunocompromised because it is very difficult to exclude the possibility of a true positive [51]. Furthermore, next-generation sequencing does not distinguish commensal or colonizing organisms from pathogenic organisms, so the results must be interpreted based on the clinical picture [52,53].

## 10. Management

The treatment of candidal infections varies considerably and is based on the anatomic location of infection, the patient’s underlying disease and immune status, the species of *Candida*, and in some cases, the in vitro susceptibility of the different *Candida* species. In 2016, the Infectious Disease Society of America and the Mycosis Study Group published guidelines for the treatment of candidiasis [54] (Table 6). Several factors are involved in treating candidemia and IC; these include removing the focus of infection (catheter, abscess), removing or decreasing immunosuppression, restoring immune function, initiating early and appropriate antifungal therapy, and assessing for metastatic foci of infection (endophthalmitis) [4,22].

Although controversial, the need for an ophthalmologic exam is still recommended according to the Infectious Disease Society of America/Mycosis Study Group candidiasis guidelines [54]. In a recently published study by Lehman et al., the investigators found an ocular candidiasis incidence of 6% of patients [55]. In addition, they also found the highest sensitivity of a positive finding when the exams were performed at >7 days from the positive blood culture.

The management of candidemia and IC in the ICU poses certain challenges. The presentation of IC in the ICU is frequently non-specific, the results of diagnostic tests are not readily available, and critically ill patients at the ICU are at risk of further rapid deterioration. These factors result in a common practice of empirically starting antifungal treatment. There are several studies that have shown that delays in the initiation of antifungal therapy translates to increased mortality among ICU patients. However, the results of randomized clinical trials have failed to demonstrate a clear benefit of pre-emptive empiric antifungal therapy [34,56,57]. The EMPIRICUS trial failed to demonstrate improved fungal infection-free survival among 251 non-neutropenic, critically ill patients with multiple *Candida* colonization, multiorgan failure, and antibiotic exposure, who were randomized to either empiric micafungin or placebo. On day 28, IFI-free survival was not statistically different between the micafungin and placebo groups [58]. However, there was a trend towards decreased rates of new proven IC in the micafungin group. In a separate study by Ostrosky-Zeichner et al., the multicenter, double-blinded, placebo-controlled MSG-01 trial randomized the patients at risk for IC in the ICU to either prophylactic caspofungin or placebo. No statistically significant difference was observed in terms of all-cause mortality as well as the development of proven IC [59]. A study by Knitsch et. al., INTENSE, aimed to assess the efficacy of pre-emptive antifungal therapy with micafungin vs. placebo among critically ill patients requiring surgery for intra-abdominal infections [60]. Again, no significant difference was seen in the occurrence of confirmed IC between the two arms, nor the median time to IC.

The most current IDSA guidelines provide a weak recommendation that fluconazole could be used among high-risk ICU patients for prophylaxis if the rate of invasive candidiasis in the ICU exceeds 5% [54]. The utility of empirical fluconazole for febrile patients in the ICU despite antibiotics was evaluated in a randomized, placebo-controlled trial by Schuster et al. [61]. Fluconazole failed to improve the composite outcome more than a placebo; however, admittedly, this study had several limitations.

It is also important to note that the widespread use of pre-emptive or prophylactic azole antifungal therapy in the ICU may be contributing to the selection of more resistant *Candida* species. According to a retrospective observational study from France, a significant increase in ICU-acquired *Candida glabrata* colonization was observed over the 2005–2012 study period, without a concomitant increase in mortality [62]. Another study detected a statistically significant correlation between the increase in caspofungin MICs among *C. parapsilosis* and *C. glabrata* and the increased use of caspofungin between 2007 and 2009 [63]. Other studies have shown that fluconazole-resistant *Candida* strains, for instance *C. parapsilosis*, can thrive even in the absence of azole exposure, making it unclear if the restriction of azoles is enough to prevent the expansion of azole-resistant strains [64,65]. Nonetheless, antifungal stewardship programs are an essential part in the management of invasive candidiasis in the ICU to ensure the responsible use of antifungals. This includes inappropriate antifungal therapy in the form of overusing antifungals without an appropriate indication, incorrect dosing, incorrect choice of antifungal, or drug–drug interactions; these have been reported in multiple studies and might contribute to adverse outcomes among the ICU population [66,67,68].

The limitations of culture and non-culture-based methods used to make a diagnosis of IC contribute to excessive pre-emptive utilization of antifungals. Blood cultures are currently considered a gold standard; however, they are frequently negative in intra-abdominal candidiasis. Moreover, even in candidemia, the prolonged time to positivity negatively impacts the time to initiate appropriate therapy. The initiation of antifungal therapy based on the results of non-culture-based testing represents another potential strategy of antifungal therapy in the ICU [36,69]. Rapid diagnostic tests (MALDI-TOF, multiplex PCR, peptide nucleic acid fluorescent in situ hybridization, and T2 magnetic resonance detection panels) can improve the time to appropriate antifungal therapy and decrease the inappropriate utilization of antifungals [70,71,72]. Among these tests, the T2MR detection panel is the only test currently approved by the FDA that does not require prior incubation and growth on culture media and can be performed straight on a blood sample.

The antifungals that are currently utilized in the ICU setting for the treatment of invasive candidiasis belong to three major groups: echinocandins, azoles, and polyenes. According to the IDSA/MSG guidelines, echinocandins are the preferred agent of choice to initially treat unstable critically ill patients [13,54,73] (Table 6). In addition, echinocandins are also the preferred initial option for both neutropenic and non-neutropenic patients due to their ability to penetrate candidal biofilms, broader antifungal activity, favorable safety profile, good tolerance, and fewer significant drug–drug interactions as compared to azoles [54,74]. In a meta-analysis by Andes et al., initial antifungal therapy with echinocandins resulted in significantly improved 30-day and 90-day mortality rates as compared to fluconazole [14,75]. Fluconazole was shown to be a good option for de-escalation therapy. The IDSA guidelines suggest de-escalation after 5–7 days, after ensuring clinical stability. Current ESCMID guidelines recommend switching to oral therapy after 10 days of intravenous therapy [13,54]. Fluconazole can still be considered as an initial management option for candidemia among ICU patients without septic shock/multiorgan failure and in *Candida* species with no fluconazole resistance [14,54]. Notably, the rapidly emerging pathogen *C. auris* is almost universally resistant to fluconazole, making echinocandins the first-line treatment. However, resistance to all three major groups of antifungals has been documented among *C. auris* strains [76,77,78]. If fluconazole is selected as the initial antifungal management option, a loading dose should be administered. It is important to note that testing for antifungal susceptibility should be pursued in all clinically relevant *Candida* isolates [4,22,54]. Lipid formulations of amphotericin B are suggested as an alternative in cases when echinocandins and/or azoles are not available and in infections due to multidrug-resistant *Candida* strains. It is worth mentioning that *Candida* species have an ability to form biofilms and thus produce biofilm-associated infections, such as catheter-associated infections, which necessitate removal of the central venous catheter. If, however, for various reasons the catheter cannot be removed, echinocandins or liposomal AmB should be used as the first line. Although no RCTs are available, based on smaller studies and expert opinions, these antifungals can be of benefit due to their ability to penetrate candidal biofilms [74,79]. Most guidelines do not consider amphotericin B an attractive first-line option due to its inherent nephrotoxic potential. Voriconazole is recommended as a step-down option for *C. krusei* infections [54].

De-escalation of the antifungal therapy, preferably oral, seems an appropriate mode of action in clinically stable ICU patients after the recommended duration of parenteral referred therapy and in the absence of antifungal resistance. In the open-label trial by Vazquez et al., there was no difference when anidulafungin was de-escalated to fluconazole/voriconazole after five days of treatment vs. continued parenteral therapy throughout the duration of treatment if the prespecified criteria for de-escalation were met [80]. Post hoc analysis of the AmarCAND2 study also showed no association between systemic antifungal therapy de-escalation and an increase in 28-day mortality. However, the study population included patients with not only proven but also suspected candidiasis, with almost half of the de-escalation group having proven invasive candidiasis [81]. In the post hoc analysis by Moreno-Garcia et al., early de-escalation in candidemia caused by fluconazole-susceptible strains after source control had been achieved had no negative impact on 30-day mortality in the multivariable analysis of risk factors [82]. Notably, these two studies were not designed to compare the ongoing therapy with echinocandins and the de-escalation to azoles. Despite the published data regarding the safety of de-escalation in clinically stable patients with fluconazole-susceptible strains, the actual de-escalation rates vary. In fact, studies evaluating de-escalation reported levels of de-escalation of systemic antifungal therapy in the ICU as low as 20–23% [81,82,83]. Further antifungal stewardship efforts are necessary to ensure appropriate and timely de-escalation of systemic therapy [15].

Biomarker-driven de-escalation of systemic antifungal therapy has shown to be a reliable option among the ICU population. The BDG assay alone or in combination with other tests (Mn-Ag/Mn-Ab) has a good negative predictive value, providing evidence to rule out IC and discontinue the antifungal therapy, as shown in two RCTs [16,84].

As stated previously, source control remains a key element of the successful treatment of invasive candidiasis/candidemia in the ICU. Failure to achieve source control is associated with higher 30-day and hospital mortality [54,57,85].

The rate of invasive candidiasis caused by drug-resistant *Candida* strains is increasing concern, rendering some of the actively utilized agents, such as azoles and echinocandins, relatively ineffective. Several novel antifungals are currently being studied as potential medicines to overcome current challenges in the management of invasive candidiasis in the setting of antifungal resistance. Ibrexafungerp is an oral triterpenoid and works by inhibiting (1,3)-β-D-glucan synthase, similarly to echinocandins. It demonstrates potent in vitro activity against drug-resistant Candida species, including azole-resistant *Candida* spp., echinocandin-resistant *C. glabrata,* and MDR *C. auris* [86,87]. Currently, the utility of ibrexafungerp in invasive candidiasis and as step-down therapy in candidemia, including infections caused by MDR *C. auris,* is being evaluated in CARES and FURI studies, with promising interim results [17,87].

Fosmanogepix is another novel antifungal that has demonstrated potent activity against *C. auris* [87,88,89] in a phase-2 clinical trial evaluating fosmanogepix in invasive candidiasis and candidemia due to *C. auris* [88]. In addition, it also showed promise as a first-line treatment of candidemia in non-neutropenic patients, with a favorable side-effect profile [87,88,89].

Finally, rezafungin is a once-weekly echinocandin with a broad spectrum of activity against *Candida* spp., including *C. auris* [90,91]. Rezafungin has demonstrated excellent activity in treating both IC and candidemia in the two randomized clinical trials STRIVE and RESTORE [92,93]. However, its role in the initial management of ICU patients needs to be further established [92,93]. Rezafungin is currently being used to continue echinocandin therapy in cases when the patients are clinically stable and ready for discharge. This avoids the use of central venous catheters in the OPAT setting while continuing once-weekly echinocandin treatment.

## 11. Empiric Therapy

The empiric use of antifungal agents in febrile septic patients in the ICU is widespread despite the absence of significant clinical data. Given the existing data regarding the use of early and appropriate antifungal therapy and its correlation with decreased mortality, it appears reasonable to recommend empiric therapy in selected patients. It is important to note that the use of empiric antifungal therapy in any febrile, low-risk patient is not justified.

There are several studies that may be used for the early identification of patients that are at high risk of developing IC and thus may benefit from an antifungal, specifically, the *Candida* colonization index, the Candida score, and the Ostrosky-Zeichner clinical prediction rule [94,95,96].

The *Candida* colonization index (CCI) is based on the premise that most patients with IC have been previously colonized by *Candida* species. The development of invasive candidal infections is often preceded by extensive multifocal colonization of the skin or of the mucus membranes of the gastrointestinal and urogenital tracts, and it is dependent on the degree of colonization. This is assessed using the colonization index, which has been shown to be an independent risk factor for the development of candidemia and IC [94]. A 6-month prospective cohort study was carried out in patients admitted to the surgical and neonatal intensive care units in a 1600-bed university medical center. The authors concluded that the intensity of *Candida* colonization, which was assessed by systematic screening, helped in predicting subsequent infections with identical *Candida* strains in critically ill patients. This study identified high-risk patients with *Candida* colonization and may offer an opportunity for early intervention strategies [95]. However, its true applicability in critically ill patients is still limited without the use of more accurate predictors, such as specific biomarkers. The *C**andida* colonization index remains an important way to characterize the dynamics of colonization, which increases early in patients who develop invasive candidiasis.

The *Candida* score is another method that may be used to identify high-risk patients prior to those patients developing positive blood cultures. In a large cohort of non-neutropenic, critically ill patients in whom *Candida* colonization was prospectively assessed, a “Candida score” of >2.5 (1 point for TPN, surgery, multifocal candida; 2 points for severe sepsis) accurately selected patients who would benefit from early antifungal treatment. The overall results from this study demonstrated a sensitivity of 81%, a specificity of 74%, a PPV of 16%, and an NPV of 98% [96].

The Ostrosky-Zeichner clinical prediction rule measures factors associated with the development of candidemia and IC. These factors include mechanical ventilation ≥ 48 h, systemic antibiotic use, and CVP (on any of days 1–3 of ICU admission), plus ≥ 1 of the following: any major surgery (days 7–0), pancreatitis (days 7–0), use of steroids/other immunosuppressive agents (days 7–0), use of TPN (days 1–3), or dialysis (days 1–3). The results revealed a sensitivity of 81%, a specificity of 74%, a PPV of 16%, and an NPV = 98% [97].

All these studies and the subsequent scores have limitations. These limitations include the potential for inaccurate predictions when trying to apply them to all critical patients, the reliance on clinical data that may not be readily available, the need for further validation in diverse settings, and their inability to definitively diagnose infection without confirmatory cultures. In essence, they should be used as tools to develop a risk assessment, not a diagnostic test [4,22,54].

Antifungal stewardship programs can be very valuable in initiating appropriate antifungal therapy. The combination of the “*Candida* Score” which was published by Posteraro et al. and the BDG assay can be a very useful tool to initiate early empiric therapy and discontinue antifungal therapy [97,98]. The criteria to initiate empiric or early presumptive antifungal therapy (echinocandin) in our institution include a *Candida* score of ≥3, high-risk patients, and a duration of >72 h on broad-spectrum antibiotics. If the patients meet these criteria, an antifungal is initiated after drawing a BDG assay. After 48 h, the BDG assay is repeated. If the BDG assay is positive, the antifungal is continued. However, if both BDG assays are negative, then the antifungal is discontinued. The use of this practice has helped decrease the use of empiric antifungal therapy by over 50% (Table 7 and Table 8). A randomized multicenter study evaluating this methodology is still needed for it to be a strong recommendation.

## 12. Conclusions

Infections due to *Candida* species are much more common than we think and remain a significant cause of morbidity and mortality in high-risk hosts [35]. These high rates are due to the lack of a diagnostic assay and the late initiation of appropriate antifungal therapy. Numerous challenges remain despite over 40 years of studying these infections, specifically, the highly variable and non-specific clinical presentations and the fact that no single, sensitive, universally applicable diagnostic assay is readily available. The current management of therapy needs to include having a high index of suspicion in high-risk groups, accurate knowledge of *Candida* species within each institution due to the increase in non-*albicans Candida* species, and the early initiation of appropriate antifungal therapy. Waiting for definitive proof of an infection increases the risk of a potentially fatal progression of the infection. Newer antifungal agents currently being studied may assist in the definitive therapy of these multidrug-resistant *Candida* species.

## Figures and Tables

**Table 1 jof-11-00258-t001:** Distribution of *Candida* species [4,5,6].

Candida Species	Isolation Rate
*C. albicans*	50–60%
*C. glabrata*	20–40%
*C. parapsilosis*	10–20%
*C. tropicalis*	6–12%
*C. krusei*	1–3%
*C. guillermondii*	
*C. lusitaniae*	
*C. dubliniensis*	
*C. auris*	

**Table 3 jof-11-00258-t003:** Common risk factors for Candidemia and invasive Candidiasis [4,22,23].

Breach in barrier defense	Host-related:
	Burns
	Mucositis
	GI perforation
	Pancreatitis
	Anastomotic leak
	Neutropenia
	Medical Interventions Performed:
	Central venous catheter
	TPN
	Abdominal surgery
	GI/liver/biliary surgery
	Urological intervention
	Renal replacement therapy
	Mechanical ventilation
	Urinary catheter
Antibiotic use	Broad-spectrum antibiotics > 72 h
*Candida* colonization	Multifocal *Candida* colonization of the skin or mucus membranes of the gastrointestinal and urogenital tracts
Environmental acquisition	*C. auris* transmission
Comorbidities	Bacteremia
	Sepsis
	Shock
	Cancer (hematologic > solid organ)
	Transplant (BMT > solid organ) diabetes mellitus
	Liver failure
	Renal failure
	Pulmonary diseases
	Hematological diseases
	HIV
	Neutropenia
Drugs	Corticosteroids
	Chemotherapy
	Anti-rejection/immunosuppressive
	Biologics
Genetics	SNPs at loci: CD58, LCE4A-C1, orf68 TAGAP (14)

**Table 4 jof-11-00258-t004:** Markers of invasive candidiasis [4,5,6].

Fever unresponsive to broad-spectrum antimicrobials (frequently the only manifestation);
Prolonged intravascular catheters;
Leukocytosis;
Prolonged stay in the ICU;
Septic shock with multi-organ failure;
Macronodular skin lesions (10–20%);
Candidal endophthalmitis (5–8%).

**Table 5 jof-11-00258-t005:** Non-culture- and culture-based diagnostic tests for invasive candidiasis [4,22,39,40,41].

Test	Turnaround Time	Result	Sensitivity	Specificity	Miscellaneous
Culture	1–4 days	Positive	40–70%	N/A	Permits the identification of species and in vitro susceptibility
Β-D-glucan	1–2 h	>80 ng/L	90%	80%	Non-specific for candidiasis
T2MR Candida	1 h	Positive	91%	99%	Allows the direct detection of five *Candida* spp in whole blood; limit of detection: ~1.3 CFU/mL
Molecular methods PCR; mcfDNA.	1–2 h	Positive	95%	92%	None are FDA-approved for clinical practice

**Table 6 jof-11-00258-t006:** Treatment of Candidemia and invasive Candidiasis [54].

Antifungal Therapy	*Candida* Species			
	*C. albicans* *C. parapsilosis* *C. tropicalis*	*C. glabrata*	*C. krusei*	*C. auris*
Preferred Initial Therapy	Echinocandins	Echinocandins	Echinocandins	Echinocandins
Alternative Initial Therapy	Fluconazole	Liposomal AmB	Liposomal AmB or Voriconazole	Liposomal AmB
Step-Down Therapy	Fluconazole	Voriconazole	Voriconazole	Variable depending on in vitro susceptibility

**Table 7 jof-11-00258-t007:** Antifungal stewardship guide for early presumptive therapy in high-risk patients with suspected Candidiasis: a stepwise approach [97,98].

Patient on broad-spectrum antimicrobials for >72 h; Negative blood cultures; Candida score > 3.
Consider the following: Obtaining β-D-glucan and one set of blood cultures; Initiating echinocandin.
At 48–72 h, perform the following: Repeat β-D-Glucan and one set of blood cultures.
If two sets of blood cultures are negative and two sets of β-D-glucan are negative, consider discontinuing the antifungal therapy.

**Table 8 jof-11-00258-t008:** Candida score calculation [97,98].

Candida Score Components	Points Assigned
Total Parenteral Nutrition via Central Line	1
Surgical ICU Admission	1
Multifocal Candida Colonization	1
Sepsis	2

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
