# Peer review of "Invasive Candidiasis in the Intensive Care Unit: Where Are We Now?"

_jof, 2025, doi:10.3390/jof11040258_

Round 1
Reviewer 1 Report
An update on such a relevant topic is always worthwhile and I congratulate the authors for their big effort, reflected in the vast bibliography cited and the delineated topics. However, it should highlight the most recent findings that provide or add information to previous publications. For instance, how is this review different from that by Calandra et al (reference 39)? Probably, from the 95 references cited, several of them dating from the last five years, more information could have been extracted or detailed to make the difference.
Why the title only mentions diagnosis and management when epidemiology and pathogenesis have more or less the same importance in the manuscript?
The manuscript lack of defined objectives. If describing diagnosis methods for invasive candidiasis is an objective (as one could infer from the title), the section needs to be more exhaustive, as it is the section about management.
Please state what each acronym means the first time it is mentioned, even if they are widely used (CRRT, ECMO, TPN, Mn-Ag, Mn-Ab, etc.). Non English native speakers could not be familiar with them.
L29: Should specify criteria to differentiate the three categories. Is not the same antimicrobial resistance, pathogenesis, high prevalence, etc.
L73-74: “…in neutropenic patients, the mortality approaches 100%”. Please provide a reference for this sentence.
L91: Please provide a reference for the affirmation that Clostridium difficile is a risk factor for invasive candidiasis/candidemia. Non of the three references in that paragraph mention it, and it is not frequently recognized, so it is an interesting information.
L106-107: “… many of these risk factors lack specificity and broad application…”, please provide a reference for this sentence.
L116 to 119: Please provide a reference.
L123 to 125: Please provide a reference for this affirmation, or an explanation by the authors proposing this classification into four syndromes.
L161-162: If sensitivity of blood cultures can reach 70%, I wouldn’t say it is low overall. Maybe it could be discriminated for different clinical settings, or the reported sensitivities among all the cited papers could be indicated in the manuscript. Regarding the time for species identification, MALDI-TOF MS technology should be mentioned as it is quite widely used in many clinical laboratories to identify the microorganisms immediately after the blood culture becomes positive, greatly reducing the tourn-around time for diagnosis. Also, other molecular methods can detect resistance genes from positive blood cultures.
L330 to 335: Should be under “Diagnosis” section, even if it’s going to be mentioned again to justify management aspects.
L361-362: What about lock therapy?
L386: The sentence supported by the reference 78 does not have much sense, unless the authors explain a bit more the content of the reference.
L424 to 463: All these paragraphs should be moved to the “Pathogenesis and Host Defense” section, along with the description of risk factors. Alternatively, the scores could be just mentioned explaining that they will be detailed in the “Empiric Therapy” section.
L467: Very interesting observation by the authors, which probably constitutes the greatest contribution of this manuscript to the management of invasive candidiasis.
L490: I think the sentence “…much more common than we think…” is quite subjective and not supported by the evidence shown in this manuscript.
Table 3. It would be useful to explain in the text (epidemiology section) how Candida auris is transmitted. Should “genetics” be all in capital letters? That would suggest a different importance with regards to the rest of the table files.
Table 4. Prolonged intravascular catheters and prolonged stay at ICU are not clinical manifestations but a risk factor and a consequence, respectively. This table could be modified to include all possible manifestations and the frequency. Otherwise is not useful.
Table 5. T2Candida MR is also a molecular method.
Table 6 confuses the reader as treatment rules are not straightforward and there are many considerations to do, as the authors actually do in the text. This table only refers to IDSA guidelines.
Table 7. It’s not an algorithm but a list of high risk factors and a table describing the Candida score. I think is not worth it to leave it.
Author Response
Major comments
An update on such a relevant topic is always worthwhile and I congratulate the authors for their big effort, reflected in the vast bibliography cited and the delineated topics. However, it should highlight the most recent findings that provide or add information to previous publications. For instance, how is this review different from that by Calandra et al (reference 39)? Probably, from the 95 references cited, several of them dating from the last five years, more information could have been extracted or detailed to make the difference.
Response: There are recent updates throughout the manuscript where applicable. A lot of candidiasis has not changed much in 5 yrs.
Why the title only mentions diagnosis and management when epidemiology and pathogenesis have more or less the same importance in the manuscript?
Response: Agree, title has been changed.
The manuscript lack of defined objectives. If describing diagnosis methods for invasive candidiasis is an objective (as one could infer from the title), the section needs to be more exhaustive, as it is the section about management.
Response: The objective was a general review. Title has been changed.
Detail comments
Please state what each acronym means the first time it is mentioned, even if they are widely used (CRRT, ECMO, TPN, Mn-Ag, Mn-Ab, etc.). Non English native speakers could not be familiar with them.
Response: Corrected throughout the manuscript.
L29: Should specify criteria to differentiate the three categories. Is not the same antimicrobial resistance, pathogenesis, high prevalence, etc.
Response: We have added the reference for these criteria for both CDC and WHO.
L73-74: “…in neutropenic patients, the mortality approaches 100%”. Please provide a reference for this sentence.
Response: agree, reference added.
L91: Please provide a reference for the affirmation that Clostridium difficile is a risk factor for invasive candidiasis/candidemia. Non of the three references in that paragraph mention it, and it is not frequently recognized, so it is an interesting information.
Response: Already referenced (16).
L106-107: “… many of these risk factors lack specificity and broad application…”, please provide a reference for this sentence.
Response: added the references.
L116 to 119: Please provide a reference.
Response: added references.
L123 to 125: Please provide a reference for this affirmation, or an explanation by the authors proposing this classification into four syndromes.
Response: There is no reference to this. The authors have used it to divide the different syndromes based upon manifestations and blood cultures,
L161-162: If sensitivity of blood cultures can reach 70%, I wouldn’t say it is low overall. Maybe it could be discriminated for different clinical settings, or the reported sensitivities among all the cited papers could be indicated in the manuscript. Regarding the time for species identification, MALDI-TOF MS technology should be mentioned as it is quite widely used in many clinical laboratories to identify the microorganisms immediately after the blood culture becomes positive, greatly reducing the tourn-around time for diagnosis. Also, other molecular methods can detect resistance genes from positive blood cultures.
Response: 40-70% we thought were low. Thus we left “low”. We have added the MALDI-TOF refence line 168-170.
L330 to 335: Should be under “Diagnosis” section, even if it’s going to be mentioned again to justify management aspects.
Response: Already a part of diagnosis section.
L361-362: What about lock therapy?
Response: Has not been studied, nor recommended in any guideline.
L386: The sentence supported by the reference 78 does not have much sense, unless the authors explain a bit more the content of the reference.
Response: Following paragraph supports this sentence.
L424 to 463: All these paragraphs should be moved to the “Pathogenesis and Host Defense” section, along with the description of risk factors. Alternatively, the scores could be just mentioned explaining that they will be detailed in the “Empiric Therapy” section.
Response: This is a part of empiric therapy and we feel this would be the most appropriate section to discuss.
L467: Very interesting observation by the authors, which probably constitutes the greatest contribution of this manuscript to the management of invasive candidiasis.
Response: Thank you.
L490: I think the sentence “…much more common than we think…” is quite subjective and not supported by the evidence shown in this manuscript.
Response: supported by reference 26. We have added it there.
Table 3. It would be useful to explain in the text (epidemiology section) how Candida auris is transmitted.
Response: Added response to line 63-64 and references.
Should “genetics” be all in capital letters? That would suggest a different importance with regards to the rest of the table files.
Response: Agree, changed to lower case.
Table 4. Prolonged intravascular catheters and prolonged stay at ICU are not clinical manifestations but a risk factor and a consequence, respectively. This table could be modified to include all possible manifestations and the frequency. Otherwise is not useful.
Response: Agreed, we have changed the title.
Table 5. T2Candida MR is also a molecular method.
Response: Agree.
Table 6 confuses the reader as treatment rules are not straightforward and there are many considerations to do, as the authors actually do in the text. This table only refers to IDSA guidelines.
Response: We agree, there are variable, but these are the actual recommendations from IDSA/MSGERC.
Table 7. It’s not an algorithm but a list of high risk factors and a table describing the Candida score. I think is not worth it to leave it.
Response: It has proven beneficial for many institutions. However, if the editor feels it should be deleted. Feel free to do so. We think it is a helpful guide to manage these problematic patients. Thus called a “possible approach”.
Reviewer 2 Report
This review does not add much to what is known, except that it promotes the use of the T2 Candida detection system for improving diagnosis without showing it improves outcomes or dealing with the cost of the assay.
This manuscript by Jose Vasquez and colleagues concerns the diagnosis and management of patients with invasive candidiasis in hospital ICUs. The authors review a lot of published literature, including several previously published reviews that cover the same ground, and recommendations by several professional societies. The emphasis is on problems with diagnosis in the ICU setting, risk factors Although this is a heterogeneous patient population in terms of underlying medical and surgical problems, and the conditions that require ICU care ranging from burns to myocardial infarcts, they are all lumped together by virtue of needing ICU care. This is a major limitation of many studies, and I am not certain how critical the authors have been in this review. A patient with necrotizing pancreatitis is quite different than one with COPD needing ventilation because of an RSV infection.
I have several specific comments:
1. Candidemia from a contaminated central venous catheter or TPN is very different than for instance candidemia in a neutropenic patient after BMT, both in pathogenesis and prognosis. The importance of normal PMN function in host resistance is illustrated on one hand by patients with AIDS who can have extensive mucositis from mouth to stomach yet are never fungaemic. In contrast a patient with acute leukemia who has a tiny bit of oral thrush will become fungemic. Yet neutropenia is not on Table 3 as a risk factor.
Table 4 is labelled Manifestations of Invasive Candidiasis, which implies that are due to invasive candidiasis, yet the first and third on the list are clearly risk factors not manifestations.
2. I strongly disagree with the recommendation to draw 1 set of blood cultures (line 482) as part of follow-up after treatment. Candida rarely grows in an anaerobic blood culture so that is in effect culturing 5-10 mL of blood. It would be better to inoculate 2 aerobic culture bottles.
3. On line 119 the increase in non-albicans Candida isolates is attribute to use of fluconazole. Is there an absolute increase in those fungi or is it relative? Since most of the invasive infections are due to dissemination of endogenous organisms rather than person to person spread, this would imply that these less pathogenic fungi were in competition with C. albicans. Is there any evidence for that?
4. Line 186. Diagnostic tests are not “broad spectrum”, ay more than a CBC is broad spectrum. I assume you are referring to the beta glucan assay, which measures that molecule but it is not specific for Candida or even fungi, but it is specific for beta-glucan. \
5. Line 20. Falling titers of beta-glucan is predictive but does not “result in” a good prognosis.
6. Line 327. Not prolonged, but “long” time. L332 and 333 needs to be rewritten.
7. Line 420 you may mean without “supporting” data.
8. In discussing that we rarely have cultures of “tissues” because patients are too unstable for surgery, you could indicate that interventional radiology can drain abscesses or viscus organs with relative ease and safety if ultra-sound can be used.
9. MALDI-tof is not a diagnostic test, it is a way to rapidly identify organisms that have been isolated by culture.
10. I think it is misleading to lump together central catheter infections with translocation from a mucosal surface, and post-op surgical infections as invasive candidiasis.
11. I do not think that the urinary tract is a likely focus for invasive candidiasis, although bladder colonization is common. In the absence of an indwelling catheter Candida in the urine is evidence of hematogenous dissemination to the kidneys.
Author Response
Major comments
This review does not add much to what is known, except that it promotes the use of the T2 Candida detection system for improving diagnosis without showing it improves outcomes or dealing with the cost of the assay.
Response: No response here. I think the reviewer misses the point of the entire manuscript. It is meant to be a review. The manuscript is not a publication of anything new.
Detail comments
This manuscript by Jose Vasquez and colleagues concerns the diagnosis and management of patients with invasive candidiasis in hospital ICUs. The authors review a lot of published literature, including several previously published reviews that cover the same ground, and recommendations by several professional societies. The emphasis is on problems with diagnosis in the ICU setting, risk factors Although this is a heterogeneous patient population in terms of underlying medical and surgical problems, and the conditions that require ICU care ranging from burns to myocardial infarcts, they are all lumped together by virtue of needing ICU care. This is a major limitation of many studies, and I am not certain how critical the authors have been in this review. A patient with necrotizing pancreatitis is quite different than one with COPD needing ventilation because of an RSV infection.
Response: Agree, however, invasive candidiasis is quite heterogenous as well.
I have several specific comments:
- Candidemia from a contaminated central venous catheter or TPN is very different than for instance candidemia in a neutropenic patient after BMT, both in pathogenesis and prognosis. The importance of normal PMN function in host resistance is illustrated on one hand by patients with AIDS who can have extensive mucositis from mouth to stomach yet are never fungaemic. In contrast a patient with acute leukemia who has a tiny bit of oral thrush will become fungemic. Yet neutropenia is not on Table 3 as a risk factor.
Response: Agree, an overcite. Added to Table.
Table 4 is labelled Manifestations of Invasive Candidiasis, which implies that are due to invasive candidiasis, yet the first and third on the list are clearly risk factors not manifestations.
Response: Agree, the title is changed.
- I strongly disagree with the recommendation to draw 1 set of blood cultures (line 482) as part of follow-up after treatment. Candida rarely grows in an anaerobic blood culture so that is in effect culturing 5-10 mL of blood. It would be better to inoculate 2 aerobic culture bottles.
Response: This is standard of care for all blood cultures, Furthermore, Candida can grow in an anaerobic BC.
- On line 119 the increase in non-albicans Candida isolates is attribute to use of fluconazole. Is there an absolute increase in those fungi or is it relative? Since most of the invasive infections are due to dissemination of endogenous organisms rather than person to person spread, this would imply that these less pathogenic fungi were in competition with C. albicans. Is there any evidence for that?
Response: No concrete evidence to support this.
- Line 186. Diagnostic tests are not “broad spectrum”, ay more than a CBC is broad spectrum. I assume you are referring to the beta glucan assay, which measures that molecule but it is not specific for Candida or even fungi, but it is specific for beta-glucan. \
Response: Line 202, we state that BDG is “pan fungal”.
- Line 20. Falling titers of beta-glucan is predictive but does not “result in” a good prognosis.
Response: not sure how to respond to this comment.
- Line 327. Not prolonged, but “long” time. L332 and 333 needs to be rewritten.
Response: Not sure where this citation is ??
- Line 420 you may mean without “supporting” data.
Response: No there is data. Just not supported yet by ICU trials.
- In discussing that we rarely have cultures of “tissues” because patients are too unstable for surgery, you could indicate that interventional radiology can drain abscesses or viscus organs with relative ease and safety if ultra-sound can be used.
Response: Agree, we have added that on line: 168-169.
- MALDI-tof is not a diagnostic test, it is a way to rapidly identify organisms that have been isolated by culture.
Response: Still considered a diagnostic assay.
I think it is misleading to lump together central catheter infections with translocation from a mucosal surface, and post-op surgical infections as invasive candidiasis.
Response: That is the definition per he IDSA guidelines.
11.I do not think that the urinary tract is a likely focus for invasive candidiasis, although bladder colonization is common. In the absence of an indwelling catheter Candida in the urine is evidence of hematogenous dissemination to the kidneys.
Response: Authors do not agree. Lower urinary tract colonization with Candida can lead to ascending pyelonephritis, especially in diabetics.
Round 2
Reviewer 1 Report
Some comments has been taken into account, like the title, which has been much improved, but other were answered in a superficial way. I feel the manuscript did not improve in a significant way.
The title has been modified to reflect more accurately the content of the manuscript.
I still think the authors need to mention an objective for writing this review. As I criticized in the last version and the authors acknowledged, a review does not need to cover all aspects of a topic. Indeed, the title, which I like, is now “where are now”. A short sentence like “In this review we aim to update/emphasize/highlight…., etc some aspects about Candida infections in the ICU”.
A text was added about transmission of Candida spp. from person to person without relationship with table 2. Please move the sentences and might be good to expand a bit on it.
Regarding this comment by me and the author’s response: L123 to 125, if this classification is proposed by the authors, it must be mentioned and explained why.
Please provide a reference for this affirmation, or an explanation by the authors proposing this classification into four syndromes. Response: There is no reference to this. The authors have used it to divide the different syndromes based upon manifestations and blood cultures,
Regarding this comment and response, I still don’t agree, especially considering that the reference dates from 25 years ago.
L490: I think the sentence “…much more common than we think…” is quite subjective and not supported by the evidence shown in this manuscript. Response: supported by reference 26. We have added it there.
Regarding this comment and response, I don’t say it’s not useful for patients’ management. All I say is that you put 2 tables with different information under a title that does not reflect the content. I’d change the title to “high risk patients” and, if you want, you can add a table 8 with “Candida score”. Just an opinion.
Table 7. It’s not an algorithm but a list of high risk factors and a table describing the Candida score. I think is not worth it to leave it. Response: It has proven beneficial for many institutions. However, if the editor feels it should be deleted. Feel free to do so. We think it is a helpful guide to manage these problematic patients. Thus called a “possible approach”.:
Author Response
Reviewer 1:
The title has been modified to reflect more accurately the content of the manuscript.
I still think the authors need to mention an objective for writing this review. As I criticized in the last version and the authors acknowledged, a review does not need to cover all aspects of a topic. Indeed, the title, which I like, is now “where are now”. A short sentence like “In this review we aim to update/emphasize/highlight…., etc some aspects about Candida infections in the ICU”.
Response: This is in the last line of the abstract.
A text was added about transmission of Candida spp. from person to person without relationship with table 2. Please move the sentences and might be good to expand a bit on it.
Response: Thank you, our error. Table 2 should not be there. The sentence has been expanded in lines 64-68. Added Reference 9, 10.
Regarding this comment by me and the author’s response: L123 to 125, if this classification is proposed by the authors, it must be mentioned and explained why.
Please provide a reference for this affirmation, or an explanation by the authors proposing this classification into four syndromes.
Response: Sentence re-structured and reference has been added.
L490: I think the sentence “…much more common than we think…” is quite subjective and not supported by the evidence shown in this manuscript. Response: supported by reference 26. We have added it there.
Reference: That is the only autopsy study to date. Unfortunately it is from 2000.
Regarding this comment and response, I don’t say it’s not useful for patients’ management. All I say is that you put 2 tables with different information under a title that does not reflect the content. I’d change the title to “high risk patients” and, if you want, you can add a table 8 with “Candida score”. Just an opinion.
Response:
Table 7. It’s not an algorithm but a list of high risk factors and a table describing the Candida score. I think is not worth it to leave it. Response: It has proven beneficial for many institutions. However, if the editor feels it should be deleted. Feel free to do so. We think it is a helpful guide to manage these problematic patients. Thus called a “possible approach”.:
Response: Agree, it is not an algorithm. The name of the table has been changed. Authors still think it is a useful guide that is used across the US and Europe in the management of patients with suspected invasive candidiasis.
Round 3
Reviewer 1 Report
Please, see attached file.
Please, see attached file.

Author Response
Thank you for reviewing the manuscript and providing very useful information. The authors have been able to address the reviewer concerns as depicted below.
- Reviewer response: I assume the authors are aware that EVERYTHING in the abstract must be present in the text of the manuscript. That is why is an abstract (of the text).
Response: Not sure what the reviewer means by the statement? The abstract is essentially a mini-paper of the full manuscript. We think it is a concise version of the paper.
- Reviewer response: Reference 25 is not available for this reviewer and reference 26 does not support the sentence. Moreover, both references are more than 20 years old.
Response: Agree both references are over 20 yrs old. However, as stated in the manuscript, this is merely a suggestion by Rex et al and by Ostrosky-Zeichner et al. in an attempt to describe the possible candidal syndromes in clinical practice. This has been the foundation for the clinical management of uncomplicated candidemia vs disseminated candidiasis. Thus the statement “It has been suggested by Rex et al…”.
- L490: I think the sentence “…much more common than we think…” is quite subjective and not supported by the evidence shown in this manuscript.
Author Response: supported by reference 26. We have added it there. Reference: That is the only autopsy study to date. Unfortunately it is from 2000.
Reviewer response: Not supported by the reference since, as the authors acknowledge, it's 25 years old, it's unique and the authors can not assume that anybody knows how common invasive candidiasis is. For sure, if you look for other references you will draw the opposite conclusion.
Response: Although the reference is 25 yrs old, there are no other studies specifically looking at causes of death in the ICU using autopsy data and invasive candidiasis in the ICU. We have re-reviewed the literature to date and have not found any further studies. In fact, the blood culture positivity rates reveal that only 40-70% of candidiasis have positive blood cultures. This is stated in the prior sentence with references (28, 29). In addition, we have added the date and author of the manuscript to clarify for the readers that it is a 2000 reference.
- Reviewer response: Again, if you want to keep a table with the Candida score, is has to be a different table or a table 7 footnote.
Response: Agree, we have added a separate table for the Candida score. In addition, we have re-structured Table 7 to be easier to use.